# Challenges in Diagnosing the Course of the Lingual Nerve for Clinical Practice and Research

**DOI:** 10.3390/diagnostics15131609

**Published:** 2025-06-25

**Authors:** Wei Cheong Ngeow, Hui Wen Tay, Krishan Sarna, Chia Wei Cheah, Mary Raj, Surendra Kumar Acharya, Zhong Zheng Koo, Mang Chek Wey

**Affiliations:** 1Faculty of Dentistry, Universiti Malaya, Kuala Lumpur 50603, Malaysia; ngeowy@um.edu.my (W.C.N.);; 2Department of Oral and Maxillofacial Surgery, Oral Pathology and Oral Medicine, Division of Oral and Maxillofacial Surgery, Faculty of Health Sciences, University of Nairobi, Nairobi 10110, Kenya; krishnasarna@gmail.com; 3Oral and Maxillofacial Surgery, Faculty of Dentistry, The University of Hong Kong, Hong Kong SAR, China; 4Private Practice, Petaling Jaya 43300, Malaysia; kzz_94@hotmail.com

**Keywords:** lingual nerve, course, clinical implication, anatomy

## Abstract

The accurate identification and protection of the lingual nerve during oral surgery are critical to avoid complications such as a loss of taste or sensation and chronic pain. While numerous studies have described the nerve’s anatomy and injury outcomes, no consensus exists on the optimal method to trace its full course. This narrative review systematically examined the literature from 2010 to 2024, using databases like PubMed, MEDLINE, Embase, and Google Scholar. Keywords included “Lingual nerve,” “Course,” “Anatomy,” and “Clinical implications,” combined with Boolean operators. Studies were selected based on defined criteria, and findings were synthesized to highlight key challenges in diagnosing the nerve’s path. This review identifies difficulties at multiple anatomical sites: the foramen ovale, infratemporal fossa, pterygomandibular space, third molar and retromolar regions, premolar/molar areas, floor of the mouth, and anterior gingiva and tongue. Lingual nerve injury, especially during lower third molar surgeries, remains a major concern, often exacerbated by factors like patient age, unerupted teeth, and lingual surgical approaches. Effective prevention hinges on precise anatomical knowledge and meticulous surgical technique. Microsurgical repair remains the primary treatment but often yields unpredictable outcomes. Emerging regenerative therapies show early promise but require further clinical validation. Imaging tools such as magnetic resonance imaging (MRI) and ultrasound may enhance diagnostic accuracy and surgical planning; however, each has limitations in everyday practice. Ultimately, early identification, careful surgical handling, and appropriate imaging support are vital for improving patient outcomes and minimizing the risks of lingual nerve injury.

## 1. Introduction

The lingual nerve (LN) arises from the posterior trunk of the mandibular nerve (V3) within the infratemporal fossa. It primarily provides sensory innervation to the mucosa covering the floor of the mouth, the lingual gingiva of the mandible, and the anterior portion of the tongue, excluding the region associated with the circumvallate papillae [1]. Additionally, the LN serves as a conduit for postganglionic parasympathetic fibers that arise from the submandibular ganglion, facilitating the secretomotor supply to the sublingual and anterior lingual glands [2,3]. It may be injured in oral and maxillofacial surgery, periodontal surgery, and bone graft and implant surgical procedures, resulting in compromised quality of life for patients [1].

The LN begins by branching off the posterior trunk of the mandibular nerve which also gives off the inferior alveolar nerve (IAN) and receives the chorda tympani, below the foramen ovale, a site where surgeons used to perform surgery/inject chemicals to control trigeminal neuralgia [4,5]. Following this, it usually courses between the tensor veli palatini and the lateral pterygoid muscles, but sometimes passes through the latter muscle as well [6]. This is a site where it has been reported to be affected by traumatic temporomandibular/condylar issues [1]. There is also speculation that it may be entrapped by muscles in this region. Furthermore, a case of entrapment within the ossified pterygospinous and pterygoalar ligaments has also been reported [7].

Thereafter, the LN lies on the deep surface of the lateral pterygoid muscle and enters the pterygomandibular space along the lateral surface of the medial pterygoid muscle. The maxillary artery crosses the LN laterally and is in direct contact at the level of the sigmoid notch [8]. This is a landmark that surgeons use when performing bilateral sagittal split osteotomy for orthognathic surgery. The LN has been reported to form communicating branches with the auriculotemporal, inferior alveolar or the mylohyoid nerves in the infratemporal fossa (Figure 1) [9]. This may result in unintended anesthesia during inferior alveolar nerve and lingual nerve (IAN-LN) blocks [10].

The LN then proceeds anteriorly and inferiorly to the medial pterygoid muscle, to gradually course to the mandibular ramus until it is in close relationship with the medial surface of this bone. This is a common site for the horizontal cut in bilateral sagittal split osteotomy in orthognathic surgery and may be injured during this procedure. It courses within the boundaries of the pterygomandibular space where the medial pterygoid muscle and the medial surface of the mandible make up its medial and lateral borders, respectively [11]. The spheno-mandibular ligament is also positioned lateral to the LN (Figure 2). This is a common site at which to deposit a local anesthetic agent for inferior alveolar and lingual nerve blocks, and may be a site of corticosteroid injection for the control of trigeminal neuralgia [4].

At the anterior edge of the pterygomandibular space, the LN passes below the mandibular attachment of the superior pharyngeal constrictor, immediately below the lower end of the pterygomandibular ligament, to course on the periosteum along the medial surface of the mandible (Figure 2). It is covered by the gingival mucoperiosteum as it courses opposite to the roots of the third molar [2,8]. The mandibular molar and retromolar regions are locations where the lingual nerve is closely related to the lingual cortex of the mandible; this is a common site for third molar surgery and the vertical cut in bilateral sagittal split osteotomy [12].

In the molar region further anteriorly, a gingival branch stemming from the LN may traverse horizontally from the medial aspect of the mandibular cortex near the retromolar pad toward the mesial area of the mandibular first molar and second premolar [12,13,14,15,16]. This branch innervates the periosteum, gingiva, and overlying mucosa of the medial alveolar process. Starting at the third molar, the lingual nerve curves away from the lingual cortex and proceeds toward the tongue in the regions corresponding to the first and second molars. This is the site for periodontal surgery and, more recently, has been gaining more attention for vertical bone augmentation procedures (Figure 3). Incidents of injuries to the LN have been reported at this site [17,18]. An implant being placed too lingually is another cause for lingual nerve injury [19].

The submandibular gland is situated at the same site where the LN bends toward the tongue. More posteriorly, it is attached to the upper pole of the submandibular gland just behind the point where the duct exits the gland. It then lies first medial to the submandibular duct and then above it for a short distance [20]. Typically, the LN courses lateral to the submandibular duct and then loops beneath it before ascending upward and forward along its medial aspect to reach the hyoglossus muscle [14,21]. However, variations have been described wherein it crosses superior to the duct [8,22,23]. This is a common site for submandibular duct/calculi and/or sublingual gland removal surgery, and may also be affected by submandibular space infection.

The LN continues on its path in an anterior and medial direction along the upper surface of the mylohyoid muscle, which forms the floor of the oral cavity. It then crosses over the styloglossus and travels along the outer surface of the hyoglossus muscle deep to the submandibular gland. Throughout this course, the nerve remains beneath the mucosa of the gingivolingual sulcus [24]. This loose mucosa gives rise to the lingual frenulum, where the submandibular duct opens. Submandibular duct exploration/calculi removal and/or ranula removal surgery may be performed here. The LN then turns around the submandibular duct at the anterior border of the hyoglossus muscle to ascend into the body of the tongue just medial to the duct and hidden under the deep lingual vein. This site is medial to the inferior longitudinal muscle of the tongue (Figure 4) [25,26]. It is closely related to the upper pole of the submandibular gland and gives off nerve branches to the submandibular ganglion that is located adjacent to the gland. The postsynaptic nerve leaves this ganglion to innervate both the submandibular and sublingual glands to secrete watery saliva [24].

Before crossing the submandibular duct, however, the LN lies between the duct and the posterior sublingual gland [20]. Besides having an intimate relationship with the LN, this posterior end of the sublingual gland also has an intimate relationship with the deep process of the submandibular gland [27,28,29]. After passing below the submandibular duct (where applicable), the LN courses deep to the sublingual gland on the genioglossus muscle [25]. A thick layer of cellulose–adipose tissue separates the medial surface of the sublingual gland from the muscles of the tongue. This is the plane to look for in the event of surgery being needed to remove the sublingual gland. However, as the terminal branches of the LN also penetrate this tissue, care must be taken during dissection (Figure 4). A gingival branch courses between the sublingual gland and the mandible to innervate the gingival mucosa and adjoining sulcus, with additional branches communicating with the mylohyoid nerve. At the floor of the mouth in this region, submental intubation may be performed to facilitate open reduction and internal reduction of fractured jawbones [30].

The LN enters the tongue at the mid-region of its lateral margin, situated on the lateral aspect of the hyoglossus muscle [25,26,31]. At this point, it penetrates the ventral lingual mucosa just anterior to the circumvallate papillae and begins branching near the anterior border of the hyoglossus muscle [25,32]. Its terminal configuration presents in one of two morphological patterns: either as a single main trunk or as two distinct primary trunks [25,26]. Before this division, the nerve typically establishes communication with either the medial or lateral branch of the hypoglossal nerve along the anterior margin of the hyoglossus muscle (Figure 4) [2,25,26,33,34].

Following the description of the lingual nerve’s (LN) anatomical pathway, this narrative review explores various approaches used to identify its origin, trajectory, and branching pattern from the foramen ovale to its terminal distribution in the tongue, floor of the mouth, and gingiva, as well as the challenges encountered in this process. The clinical implications will also be discussed.

## 2. Materials and Methods

The present narrative review was conducted to systematically analyze the existing literature on the challenges in diagnosing the course of the lingual nerve for clinical practice.

### 2.1. Search Strategy

A systematic literature search was conducted in PubMed, MEDLINE, Embase, and Google Scholar to identify relevant studies published between January 2010 and December 2024. The search strategy employed a combination of MeSH terms and keywords related to the lingual nerve, utilizing Boolean operators (“AND” and “OR”) to refine the search results.

The primary search terms included

“Lingual nerve” AND “Course”;“Lingual nerve” AND “Anatomy”;“Lingual nerve” AND “Clinical implications”.

All studies discussing the anatomical course and challenges in diagnosing the lingual nerve were considered for inclusion.

Inclusion Criteria:Studies focusing on the course and anatomy of the lingual nerve;Studies published in English;Original research, systematic reviews, and meta-analyses.

Exclusion Criteria:Case reports, letters to editors, and non-peer-reviewed articles;Studies focusing on surgical techniques without anatomical discussion;Articles not available in full text.

### 2.2. Data Extraction and Analysis

Three hundred and seventy two articles were retrieved. Two independent reviewers screened the articles for relevance based on title, abstract, and full-text assessment. One hundred and twelve out of 372 articles were included (Figure 5). Data extraction was performed for key parameters, including:Challenges in locating LN during clinical and surgical procedures;Methods used for anatomical guidance and diagnosis.

When discrepancies/disagreement arose, this was resolved through discussion or consultation with a third researcher. The extracted data were integrated to identify challenges in diagnosing the course of the lingual nerve.

## 3. Results

### 3.1. Challenges Associated with Landmark Identification at the Foramen Ovale

Historically, chemical neurolysis with alcohol was a commonly performed procedure for the management of trigeminal neuralgia, targeting various branches of the trigeminal nerve, including the lingual nerve [35,36]. Extracranial injection of alcohol into the trigeminal nerve as it exits the foramina ovale and rotundum was first introduced by Pitres and Verger from France in 1902 and Schlosser from Germany in 1903. Subsequent refinements of the technique by Ostwald, Levy, and Badoin led to its widespread adoption among American surgeons, who reported analgesic durations comparable to those achieved with peripheral neurectomy [37,38].

There are two main techniques for the delivery of peripheral alcohol injections. The first involves transcutaneous injection at the skull base, targeting the trigeminal nerve as it emerges from the foramen ovale or foramen rotundum. The second technique targets specific branches of the trigeminal nerve by delivering injections more distally at points where the nerve exits foramina in the face such as the supraorbital, infraorbital, inferior alveolar, and mental foramina. Injection of 1–2 mL of absolute ethanol is typically performed, without direct visualization. Mckenzie described that the only reliable indicator of correct needle placement was the ability to elicit a paroxysmal, radiating pain response from the patient, indicating direct nerve contact [39]. However, this blind approach carries substantial risks. It has been reported that deep injection of the trigeminal ganglion through the foramen ovale can cause alcohol to enter the subarachnoid space. To reduce these complications, Sweet in 1950 began taking radiographs to ascertain the needle position prior to injection [40]. However, this approach only provides an approximation of the location of the trigeminal nerve based on anatomical landmarks without a significant degree of accuracy. With recent advances in neuroimaging modalities and minimally invasive techniques, the use of alcohol neurolysis has since been abandoned.

Modern technology has facilitated safer and more precise means for treating trigeminal neuralgia in the foramen ovale region. The use of virtual navigation with multi-modality image fusion [41] allows for real-time visualization of adjacent anatomical structures, enhances procedural accuracy, and minimizes complications. Good outcomes were reported when this technique was used for percutaneous rhizotomy of the trigeminal nerve [42], percutaneous balloon compression [43], and foramen ovale cannulation [44]. However, many of these studies were limited by a small sample size or were performed on cadaveric specimens. Additional clinical studies are needed to improve our understanding of this subject.

### 3.2. Challenges in Diagnosing the Lingual Nerve in the Infratemporal Fossa Region

The LN has been observed as passing through the infratemporal fossa on its way inferiorly to the pterygomandibular space, after branching from the mandibular division of the trigeminal nerve (CNV3). There are reported variabilities in the location of bifurcation of the IAN and LN (Figure 6). The clinical significance lies in mandibular condylar neck or subcondylar fractures, where the LN carries a higher risk of damage by medially dislocated sharp bone fragments if the fracture is near the site of bifurcation [45]. Moreover, cadaveric studies show that it can become impinged or entrapped by muscles in this region, resulting in neuropathy [6,46,47]. Potential entrapment sites of the lingual nerve within the infratemporal fossa include the following: (a) ossified pterygospinous or pterygoalar ligaments—whether partially or fully ossified—near the cranial base; (b) the large lamina of the lateral plate of the pterygoid process; and (c) the medial fibers located in the anterior part of the lateral pterygoid muscle [46]. The diagnosis is, however, usually derived by the exclusion of other causes such as temporomandibular joint dysfunction syndrome and temporomandibular disk or condylar fracture displacement [1]. Since this proposal was made about 15 years ago, no update has been provided in the literature. It is important that a magnetic resonance imaging (MRI) scan is undertaken in order to ascertain if LN entrapment does exist in patients with lingual nerve neuropathy. Given its relatively large size, MRI should be able to pick up any anomaly that affects the LN in this region [48].

### 3.3. Challenges in Diagnosing the Lingual Nerve in the Pteryomandibular Space

IAN and LN (IAN-LN) regional blocks are performed blindly in the pterygomandibular space using intraoral landmark techniques. These injections have a risk of unanticipated nerve and arterial injury, leading to a higher failure rate besides unintended complications [49]. There is a paucity of data in the literature on the incidence of LN damage caused by local anesthesia [50,51], with a relatively low incidence of 0.15–3.65% [52,53]. As the risk of LN injury during an IAN block is extremely low, this potential risk is usually underestimated by surgeons and patients are not as well-informed. The mechanism of injury is either by direct mechanical needle penetration of the nerve trunk; or indirect pressure by intraneural hematoma due to intraneural vessel rupture or the neurotoxicity of anesthetic agents [54]. The latter may see a gradual return of nerve function to normal, usually taking several weeks for remyelination [55]. If nerve injury does occur, the LN has been reported to sustain more unintended intraneural complications compared to the IAN, with neurotoxicity from 4 percent prilocaine and 4 percent articaine of 7.3 and 3.6 times, respectively [50,56]. Pogrel et al. attributed the higher incidence of LN involvement in this region to it having a unifascicular nerve based on a cadaveric study that was conducted [57]. The unifascicular pattern was more common at the lingula level (site of injection) than the multifascicular pattern seen in lingual to the mandibular third molar region. An earlier study reported a favorable outcome after direct damage to the LN, with 17 out of 18 patients reporting that their sensation recovered after 6 months, while the remaining patient (0.008%) experienced a slight sensory diminution of the tongue, even after one year [53]. In contrast, unfortunately, more recent studies stated that these LN injuries were persistent without spontaneous improvement in neurosensory and/or gustatory function, perhaps as a result of neurotoxicity instead of direct damage. It has been suggested that the utilization of a small-diameter 27-gauge needle, minimizing the number of injections, and a shallower depth of needle insertion [58] anterior to the mandibular foramen during an IAN block may help to further reduce the risk and severity of LN injury [54]. Nevertheless, this is still a blind procedure.

The anatomical variability of the LN including its size and location renders difficult the prediction and prevention of such injury. Hence, the ability to “see” where an injection is given can be helpful. Watanabe et al. [59] developed the “inferior alveolar nerve block mandibular angle approach (IANB-MA)” that enabled the performance of IAN-LN blocks guided by ultrasound and tested it successfully on four cadavers. An ultrasound probe was placed on the lower border of the mandible while a needle was advanced medial to the mandible. Through the use of blue acrylic ink injection, its spread was evaluated by dissection. They reported that the LN was consistently dyed, and when applied clinically, this method reduced the pain suffered by three patients with trigeminal neuralgia or tongue or jaw pain. They concluded that it may be a good approach to control pain in comparison to other conventional approaches.

Mandibular orthognathic surgery, specifically sagittal split ramus osteotomy (SSRO), carries a significant risk of lingual nerve injury due to the nerve’s proximity to the osteotomy cut-line. The overall prevalence of lingual sensory impairment has been estimated to be 0.1% [60]. When an LN injury occurs in association with sagittal ramus osteotomy, it may happen at the time of incision placement, during subperiosteal dissection along the medial ramus, or when using a rotary drill or reciprocating saw during medial ramus osteotomy [60,61]. LN injury is more common during mucosal incision and subperiosteal dissection along the medial ramus because of the contiguity and parallel course along the inner surface of the mandible [62]. At the level of the mandibular canal, the LN is approximately 1 cm in front of the lingula and can be injured during the cortical cut of the medial ramus. In addition, injury may occur less commonly during fixation, electrocautery, wound closure, or the retraction/compression of the mandible for repositioning [61]. Overzealous lingual retraction or the placement of long screws that perforate the lingual cortex may cause neurosensory disturbances [63]. As the mechanism of injury is commonly related to the traction of the nerve, injuries are mostly transient [64]. All these procedures are dependent on the visibility of the relevant site, as sometimes they were conducted blindly based on the estimation of surgical planning. If surgeons can clearly view what is being carried out, or are able to deflect the lingual nerve away, the risk of LN injury will reduce tremendously. Perhaps the use of a fiber-optic light in this tight place may help.

### 3.4. Challenges in Diagnosing the Lingual Nerve in the Third Molar and Retromolar Regions

The LN is susceptible to injury during surgical third molar removals, particularly when the lingual split technique is utilized [65]. This vulnerability is attributed to the nerve’s close anatomical proximity to the lingual mandibular cortical plate in the molar and retromolar regions, where direct contact with the lingual cortex has been observed in approximately 20% to 62% of cases [21,66,67,68,69]. Furthermore, in 4.6% to 21.0% of cases, the nerve may be positioned at or above the alveolar crest [21,66,67,69,70,71], and in 0.15% to 1.5% of instances, it may be located within the retromolar pad area [67,69].

Although the buccal approach is commonly adopted to remove the mandibular third molar, there would be occasions wherein dentists/surgeons need to work at the lingual plate, e.g., when the tooth is distolingually oriented or when retrieving a lingually displaced mandibular third molar (Figure 7). On these occasions, the dentists/surgeons would need to protect the lingual nerve from iatrogenic injury, but the process to identify and protect this nerve itself would cause injury if its presence is not identifiable [72].

Besides ultrasound (US), magnetic resonance imaging (MRI), first used by Miloro et al. in 1997 [68], is a promising imaging modality that could become part of routine clinical practice [73]. Many researchers confirm its validity in visualizing the LN in healthy patients for pre-treatment planning [74,75,76,77,78,79,80] and in patients with nerve lesions, injury, or neuropathy [81,82,83]. Although promising, Fujii et al. reported difficulty in observing smaller nerves, such as the mylohyoid nerve [76]. So lingual nerve branches with small diameters would not be detectable.

### 3.5. Challenges in Diagnosing the Lingual Nerve in the Premolar/Molar Region

As stated earlier, the lingual nerve may give off a lingual gingival branch [13,14,15,16]. This branch is usually located superior to the main trunk, as evidenced by live and cadaveric dissections [15]. Extensive flap elevation such as that shown in Figure 3 may injure these branches. Kocabiyik et al. [16] suggested that neurosensory disturbance on the lingual gingival tissue after the difficult extraction of impacted lower third molars may be due to the manipulation of the mucoperiosteal flap containing this branch, which in most cases can be misdiagnosed as paresthesia of the main lingual nerve. It is challenging clinically to determine the presence of this nerve, so the potential of ultrasound and MRI, as described above, should be explored. They have not been observed using ultrasound or MRI scans, perhaps because this was not the aim of these studies or they were too small to be captured [27,84].

### 3.6. Challenges in Diagnosing the Lingual Nerve at the Floor of the Mouth

Upon reaching the lingual plate near the mandibular third molar, the LN travels anteromedially above the surface of the mylohyoid, moving its path in between the sublingual and submandibular glands, crossing the styloglossus and then running its course on the hyoglossus muscle at the floor of the mouth before terminating at the genioglossus muscle of the tongue, providing general sensation to these regions [22,25,85,86]. Current knowledge of the position of and relationship between the LN and the submandibular duct is obtained as an outcome of cadaveric dissection mainly or clinical observation during surgery. No consistent position has been described at the point where the LN intersects the submandibular duct, although one study reported that this often occurs around the region of the premolars [8], while another study reported said landmark to be in the interproximal space between the lower first and second molars on the mandibular arch [86]. In the former cases, the distance between the alveolar plate, at the level of the third molar region, and the sublingual region of the intersection was found to be 22.6 mm [8]. Al-Amery et al. instead found that the mean distance of overlap between these two structures was 6.92 mm, happening at the region between the mandibular first and second molar teeth. They also found that the mean distance between the LN and alveolar ridge was 12.36 mm and the mean distance between the LN and the lower border of the mandible was 12.03 mm [22].

A cadaveric study performed almost half a century ago reported that this position was inconsistent and can be anywhere from the distal of the mandibular second premolar to the retromolar trigone. Fifty-five percent of this intersection was reported to occur at the mandibular third molar tooth or behind it [27]. Occasionally, the submandibular duct runs deep in the floor of the mouth, with no relationship with the LN. This accounted for 7.7–11.8% of samples in different cadaveric studies conducted by Hölzle and Wolf and Al-Amery et al. [21,22].

It is imperative to know the course and position of the lingual nerve in the submandibular region during surgical procedures involving the floor of the mouth like sialolith removal, tumor removal, or biopsy. The LN is reported to either loop below the Wharton’s duct or cross above or run parallel to it, as described earlier [8,23,69]. Currently, surgeons are dependent on locating the LN from the submandibular gland/duct and slowly dissecting around this nerve should they need to perform surgery on this region. The use of ultrasound may ease surgery [87], but there is a learning curve that surgeons experience when applying this in the operation theater. The use of soft tissue window in computed tomography for LN detection has proven to be insubstantial because of poor resolution and artifact interference; cone-beam computed tomography too cannot provide accurate soft tissue information, limiting the direct visualization of the LN [88]. Other alternative methods of locating the LN from the submandibular gland or duct such as MRI have been introduced [73,87,88] but are not routinely used.

Lastly, submental intubation is an anesthetic procedure conducted at the floor of the mouth blindly, with the risk of causing injury to the LN [30]. There have been no reports/attempts to overcome this in the current literature. Perhaps ultrasound may play a role in confirming the presence of the LN.

### 3.7. Challenges in Diagnosing the Lingual Nerve at the Gingiva Anteriorly and Tongue

The LN gives off branches that communicate with the mylohyoid nerve, sometimes termed as the “mylohyoid or sublingual curl” [13]. The prevalence of such communication has been reported to range from 12.5% to 33.3% [13]. Besides providing sensation to the lower anterior teeth and chin via the mylohyoid nerve, such communication may assist in recovery from LN injury if communication occurs in close relation to this region [89]. The presence of these communications may explain sensory recovery in patients undergoing genioplasty. Fazan et al. are of the opinion that the mylohyoid nerve would be contributing to the sensory innervation of the tongue in such an event [89]. However, these findings were purely anatomical findings from cadaveric dissection or observations during floor-of-the-mouth surgery. As the ultrasound study by Barootchi et al. did not report observing any such communication [84], it would be interesting to see whether an ultrasound study can be performed to determine the difference in the prevalence of communication in patients with LN injury. The ability to determine the difference between traumatized and non-traumatized patients with this tool will assist dentists and surgeons in treatment planning and in predicting treatment outcomes.

The LN is also in communication with the hypoglossal nerve, with a prevalence of 40–83% [90,91]. Shinohara et al. described two patterns of communication, namely, the anterior type in the sublingual region (26.6%) and the posterior type in the submandibular region (56.7%) [90]. Iwanaga et al. instead reported that communication between the lingual and hypoglossal nerves could be observed at the anterior, middle, and posterior thirds of the tongue in all of their cadaveric specimens [92]. On the coronal plane, Fitzgerald & Law reported the presence of a medial and a lateral lingual–hypoglossal connection, with the latter being more commonly observed [91]. Their findings suggested that the lingual nerve is a proprioceptive pathway for both extrinsic and intrinsic muscles of the anterior part of the tongue. On the other hand, there is also a suggestion that the communicating branches convey motor fibers to intralingual muscles [33]. The clinical implication of these communications is limited, with two case reports of combined hypoglossal and lingual nerve palsy after orotracheal intubation for general anesthesia [93,94].

Hence the ability to diagnose the location of these nerves is of utmost importance. MRI has been reported to have difficulty in visualizing the trochlea nerve (0.3–1 mm) and abducens nerve (1 mm) due to their oblique direction and small size [48], although more recent technology allows for visualization at different parts of these small nerves (i.e., not as an entire continuous structure) [95,96]. Hence, visualizing communicating branches which may be even smaller in size is the utmost challenge of MRI.

## 4. Discussion

Lingual nerve injury remains a significant clinical complication, characterized by neurosensory disturbances and altered taste affecting the anterior two-thirds of the tongue. Patients may also experience dysesthesia and neuropathic pain which can significantly impair their quality of life, leading to anxiety, depression, and social reclusiveness [97,98]. Surgical removal of the mandibular third molar is a commonly performed procedure and has the potential to cause lingual nerve injury due to the nerve’s superficial course along the lingual cortex of the mandible and its proximity to the third molar region. Increased age, unerupted teeth, distoangular impaction, lingual flap elevation, and the lingual split technique were reported to be significant risk factors for lingual nerve injury during mandibular third molar surgery [99]. Understanding the mechanisms of nerve injury, combined with precise anatomical knowledge, is essential for clinicians to reduce the risk of iatrogenic nerve injury and to facilitate successful microsurgical repair in instances where inadvertent injury occurs.

The first nerve injury classification system was proposed by Seddon in 1947 [100]. Neurapraxia is defined as focal segmental demyelination without axonal disruption, typically caused by compression injury and resulting in temporary conduction block. In axonotmesis, there is axonal disruption and Wallerian degeneration with intact connective tissue and nerve continuity. It usually arises from a crush injury. Neurotmesis refers to complete nerve transection, resulting in complete conduction block. Sunderland revised the classification system in 1951 [101]. Type I was defined as local myelin damage, Type II as axonal damage with intact endoneurium, Type III as axon and endoneurium damage with an intact perineurium, Type IV as axon, endoneurium, and perineurium damage with an intact epineurium, and Type V as complete nerve transection. Seddon’s Neuropraxia correlates with Sunderland’s Type I category, while axonotmesis and neurotmesis correspond to Types II–IV and V, respectively.

The technique for microsurgical repair is described below (Figure 8) [102].

A buccal mucoperiosteal incision is made, extending from the external oblique ridge to the distobuccal aspect of the mandibular second molar. To achieve adequate surgical exposure, an additional incision is made along the lingual gingival margin extending from the mandibular second molar to the ipsilateral mandibular canine. Both buccal and lingual mucoperiosteal flaps, connected posteriorly at the distal aspect of the mandibular second molar, are then carefully elevated using periosteal elevators. A superficial incision is then made at the lingual periosteum and careful blunt dissection is performed to identify and expose the lingual nerve. Any observed traumatic neuroma should be surgically excised, measured, and submitted for histopathological evaluation. If the nerve is adherent to adjacent tissues, external neurolysis may be performed. Distal and proximal nerve stumps are then mobilized and prepared for anastomosis. Care should be taken to ensure that the ends meet without tension. Epineural suturing is performed with 7/0 nylon sutures, followed by wound closure. Axonal regeneration occurs at a slow rate of 1 mm/day [103].

While microsurgical repair remains the main treatment modality for managing lingual nerve injuries, recent advancements in neuroscience have shown promise in enhancing nerve regeneration and improving functional sensory recovery. In particular, emerging therapies such as the use of erythropoietin and stem cell-based interventions have demonstrated promising results in preliminary studies. In particular, the incorporation of stem cells into biomaterial-based scaffolds to increase the effectiveness of tissue-engineered nerve grafts and boost the neuro-regenerative process shows significant potential [104]. However, most of these new studies have only been trialed on animal models, and further research is necessary to ascertain clinical safety, efficacy, and long-term outcomes [105].

Despite the availability of many techniques to improve nerve recovery, the outcome of any nerve repair is not always favorable [106], with less than half (44.7%) of patients reporting sensory improvement in one study [107]. In addition to the previously mentioned method, direct perineurial suturing using a vein graft cuff and the use of indirect collagen allografts have also been employed in the management of transected lingual nerves [108]. Both techniques have demonstrated favorable results; however, from a cost-effectiveness perspective, vein graft-assisted direct repair is considered advantageous, despite potential aesthetic concerns at the donor site. Conversely, interpositional collagen allograft repair offers the benefit of significantly reduced operative time.

Thus, the best practice available would be to “see” the LN and hence avoid injuring it, unless the procedure is for tumor resection purposes. Figure 9 provides a clinical decision algorithm for when it is necessary to “see” the LN. The currently available imaging modalities for this purpose include magnetic resonance imaging (MRI) and ultrasound, as described.

MRI is costly and may not be easy to interpret by non-radiologists. Taking an MRI for the purpose of locating and assessing the lingual nerve will require a head coil or surface coil over the head to create a temporary magnetic field; the radio-wave signals will then be constructed into images [109]. Caillet et al. [48] reported that MRI is a sensitive examination technique that recognizes cranial nerves and should be the first-choice investigation method in patients with cranial nerve pathology. Its use has proven to be reliable; the lingual nerve, be it intact or injured, could be seen and delineated from surrounding tissues under different MRI sequences [74,75,76,77,78,81,82,83,109]. With accurate visualization of the course, the measurement of the LN, especially on the mandibular third molars and in the floor-of-the-mouth region, is possible under high-resolution MRI [68,77,78,88], using three protocols. Burian et al. showed that short tau inversion recovery had the best quality in locating the nerve and that double-echo steady state was best for nerve diameter measurement [77]. Standardized protocols of MRI sequences for the assessment of the lingual nerve have not yet been established [110]. Also, its use for diagnostic purposes has not been proven in routine practice. In fact, one publication that provided a pictorial overview of the cranial nerves failed to mention any branches or communicating branches of the LN [111]. It also cannot be used clinically, for example, to help perform IAN and LN nerve blocks.

Current reports indicate that MRI is best used as a diagnostic tool for third molar surgery, although this may be to a limited extent [73,79,80,110]. Nevertheless, it may be used for the diagnosis of lesions in the LN, as described by Al-Haj Husain et al. [110]. Conventional MRI was reported to have a sensitivity of 18% and specificity of 100% in detecting post-traumatic trigeminal neuropathic pain [112], while a newer method, magnetic resonance neurography, has an overall sensitivity of 38.2% and specificity of 93.5% [113]. In detecting lingual nerve injury, sensitivity and specificity are 48.6% and 96.5%, respectively, and sensitivity improves when the injury falls under more severe categories according to the Sunderland classification [113]. Lastly, Miloro et al. [68] and Al-Amery et al. [114] reported that the advantage of MRI in diagnosing lingual nerve injuries would be limited to complete transections, suggesting its limited sensitivity and specificity in nerves with subtle lesions.

Ultrasound, although cheaper, is technique-sensitive and needs a learning curve to be applied to routine clinical practice. Factors such as the operator’s experience (familiarity with ultrasonography), the equipment used (i.e., high-resolution ultrasound machines with appropriate transducers), and the anatomical region being assessed might affect its sensitivity and specificity [115]. It has been employed on various occasions, i.e., in the pterygomandibular space for an ultrasound-guided IAN-LN block, the lingual plate region for a study to measure the LN, and the floor of the mouth to locate the submandibular gland/duct. In contrast to MRI, US is simpler to use, and more and more recent studies have reviewed the feasibility of introducing it into routine dental practice [116]. For example, its use to measure gingival thickness showed that it has accuracy not significantly different than direct readings [117].

It has been suggested that ultrasound’s sensitivity and specificity might be greatly improved when used adjunctively with anatomical landmarks. However, there is no published study that provides data on the sensitivity and specificity of using ultrasound in trigeminal/lingual nerve identification. The closest information available is the meta-analysis published in 2011, using ultrasound to diagnose Carpal Tunnel Syndrome (CTS), where it showed a sensitivity and specificity of 77.6% and 86.8%, respectively [118]. Ultrasound works well when used in soft tissue unhindered by bony structures, such as in showing the course of the lingual/sublingual artery for pre-operative planning [119,120], but fares badly when evaluating structures behind bone [121], although Al-Amery et al. managed to do so in their cadaveric study [114]. So its benefit in diagnosing the LN routinely is untested, as it is hidden behind the mandible in almost half of its course. The study by Al-Amery et al. claimed to be able to scan all 12 lingual nerves with injury, suggesting 100% specificity [114].

Ultrasound is an exciting mode of imaging in the dental office as it is portable minus the risk of exposure to ionizing radiation and, at the same time, allows for repeated examinations. Its application to detecting trigeminal nerves has been reported, but mainly for the main branches. More and more publications since the 2010s advocate the use of ultrasound in dentistry, but unless a dentist/oral and maxillofacial surgeon is prepared to adopt this mode of imaging in routine use, its use may still be restricted to oral and maxillofacial imaging experts. Díaz et al. [116] reported that the operator must undergo extensive training to use ultrasound and also have good medical knowledge of anatomy, physiology, and pathology, in order to bring the best out of ultrasound imaging. The operator must also be knowledgeable in scanning techniques, have the ability to identify artifacts, and be able to troubleshoot technical issues. This is a steep learning curve that may put off some beginners.

## Figures and Tables

**Figure 1 diagnostics-15-01609-f001:**
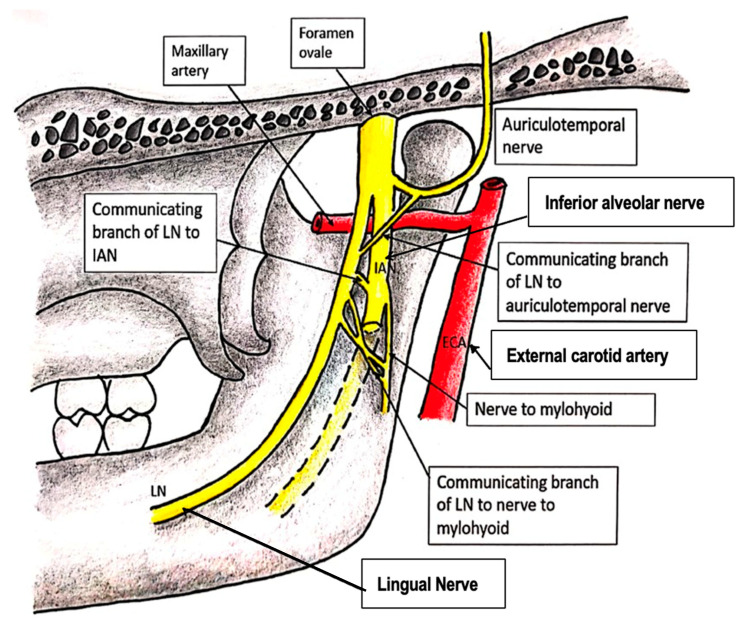
Nerves communicating with the LN in the infratemporal fossa.

**Figure 2 diagnostics-15-01609-f002:**
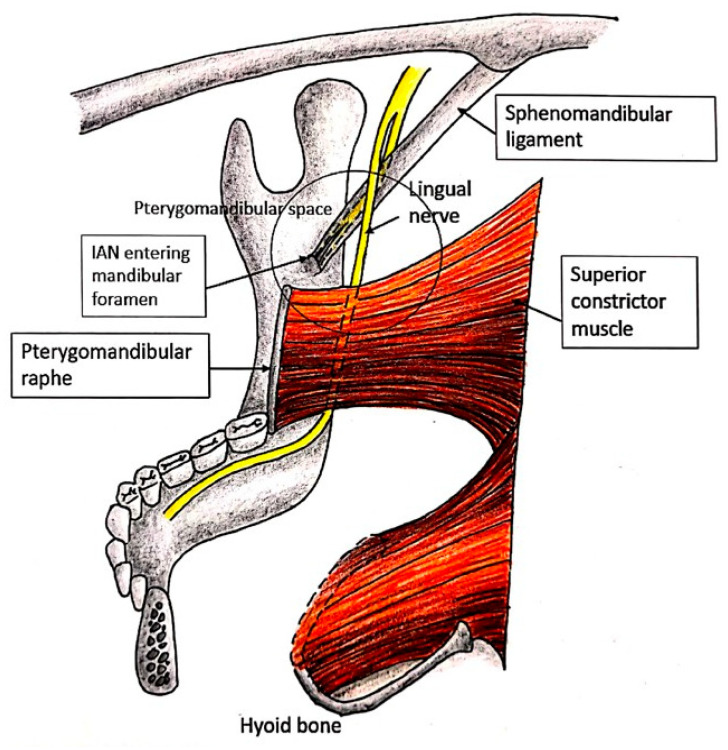
The LN in the pterygomandibular space.

**Figure 3 diagnostics-15-01609-f003:**
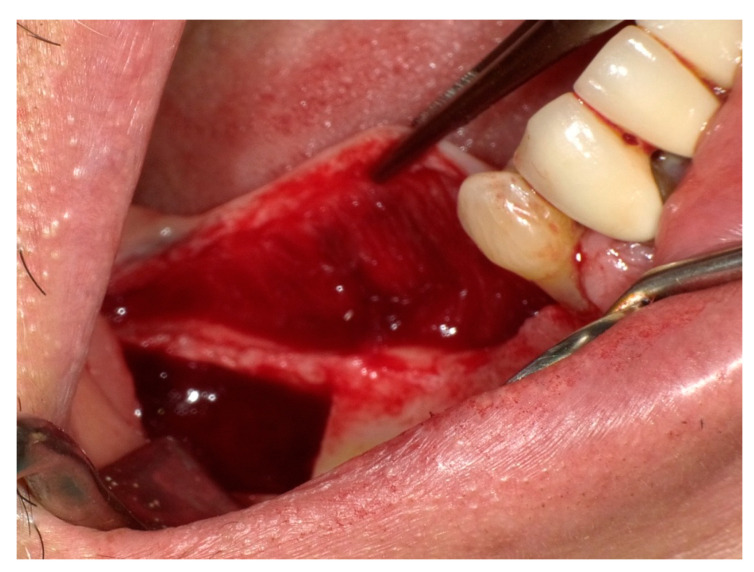
The lingual mucoperiosteal flap for vertical augmentation.

**Figure 4 diagnostics-15-01609-f004:**
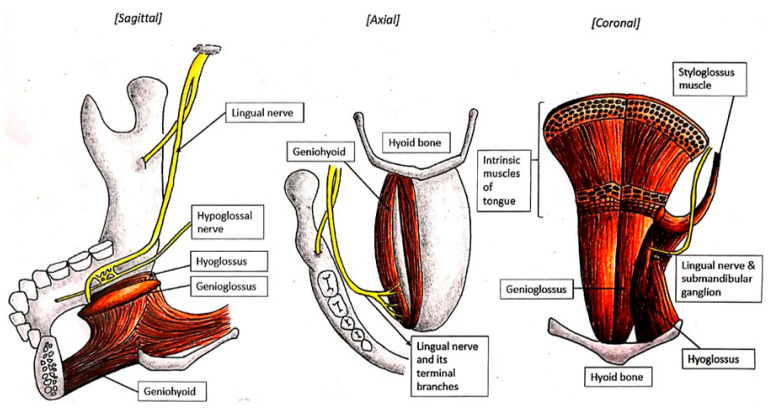
Lingual nerve innervation at the floor of the mouth.

**Figure 5 diagnostics-15-01609-f005:**
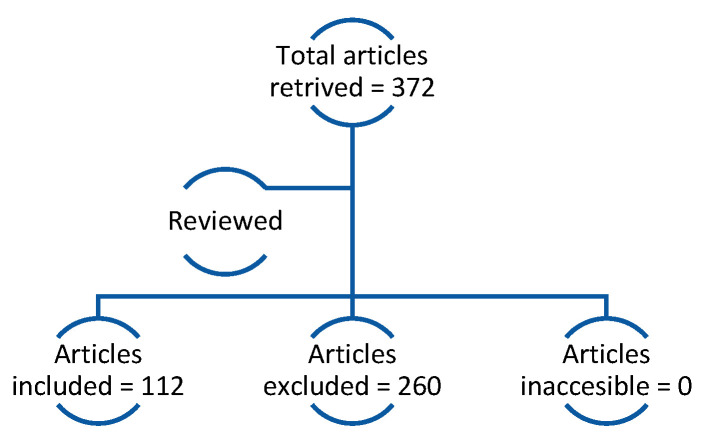
Flow chart of article search showing included, excluded, duplicate, and inaccessible articles.

**Figure 6 diagnostics-15-01609-f006:**
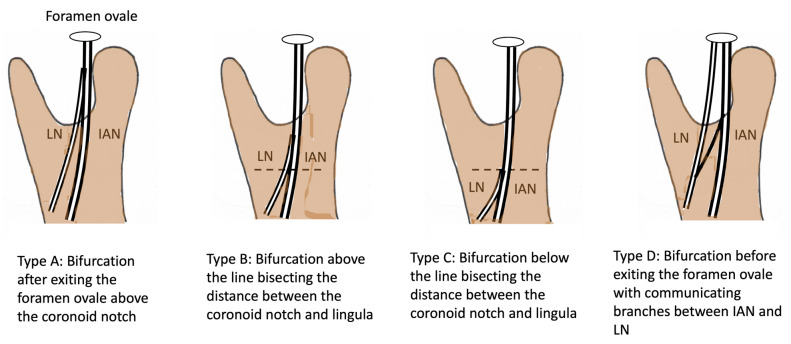
LN patterns of bifurcation at different levels beneath the foramen ovale.

**Figure 7 diagnostics-15-01609-f007:**
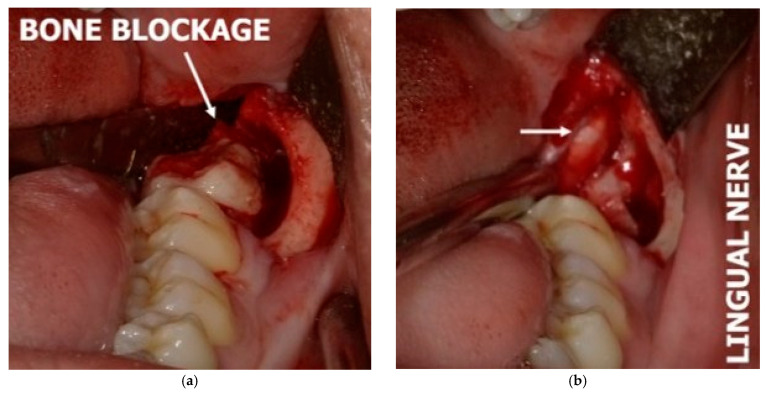
(**a**) Lingual nerve protected by a periosteal elevator; (**b**) arrows highlight the appearance of the lingual nerve observed after the removal of the periosteal elevator.

**Figure 8 diagnostics-15-01609-f008:**
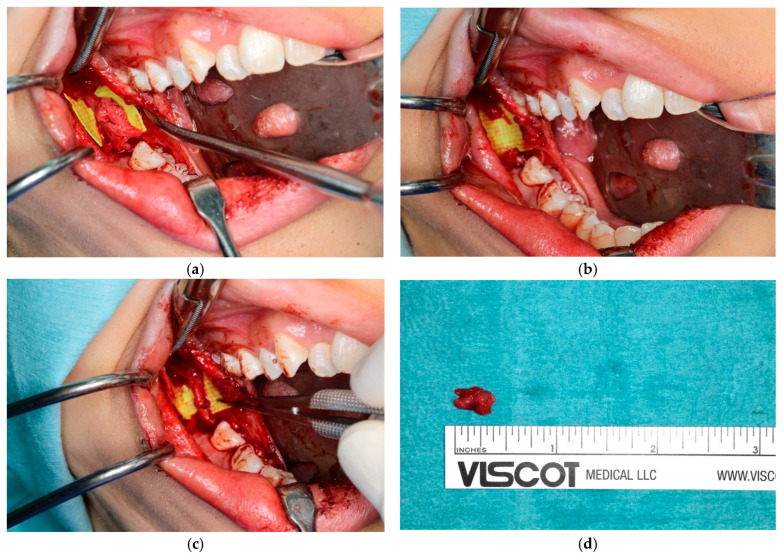
Microsurgical repair of lingual nerve: (**a**) neuroma associated with lingual nerve; (**b**) surgical excision of neuroma; (**c**) microsurgical repair of lingual nerve; (**d**) excised neuroma.

**Figure 9 diagnostics-15-01609-f009:**
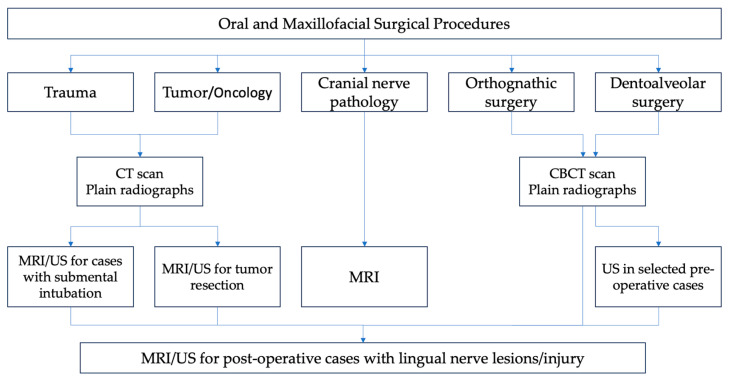
Clinical decision algorithm for pre-operative and post-operative cases based on anatomic risk factors.

## Data Availability

No new data were created.

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
