# Peer review of "Challenges in Diagnosing the Course of the Lingual Nerve for Clinical Practice and Research"

_diagnostics, 2025, doi:10.3390/diagnostics15131609_

Round 1

Reviewer 1 Report

Comments and Suggestions for Authors

I read the manuscript titled "Challenges in Diagnosing the Course of the Lingual Nerve for Clinical Practice and Research" with great interest and curiosity. I sincerely congratulate the authors on the successful execution of their work and the comprehensive review that resulted from it. As the authors have rightly pointed out, the lingual nerve is susceptible to injury in dental practice and clinical settings, which can lead to significant complications. In this regard, conducting a review on this topic is highly appropriate.

However, if your manuscript is a narrative review, please explicitly state this at the top of the manuscript instead of referring to it as a comprehensive review. Comprehensive reviews are, by definition, systematic reviews.

Additionally:

  • Please add current references to the first two sentences of the Introduction section.

  • Was Figure 1 drawn by you? If so, congratulations. However, if it was taken from a source, it cannot be used in this way—you must include an original drawing.

  • The same applies to your other figures. For images that include photographs, if patient consent was obtained, this must be clearly stated. If not, such images should be removed. For other figures, please consider the same issues I mentioned above. Otherwise, copyright for those drawings belongs to their original creators.

  • The Materials and Methods section needs to be expanded with more detail. Also, to comply with review guidelines, it would be appropriate to include a table/figure showing included, excluded, duplicate, and inaccessible articles. I recommend reviewing other similar literature reviews for guidance.

  • Your Results section is well-prepared—congratulations. However, the Discussion section should be expanded. In addition to the lingual nerve, you should include references to publications that discuss similar nerve injuries in general, thereby enriching the section. Overall, your Discussion is more superficial compared to other sections and requires further elaboration.

For your research to be eligible for publication, these issues must be fully revised. I look forward to reassessing the manuscript after the revision.

Reviewer 2 Report

Comments and Suggestions for Authors

The paper is comprehensive, well-referenced, and clinically relevant.

However, there are areas that need improvement in terms of clarity, scientific rigor, and organization.

* Although the paper claims to follow PRISMA, this is a narrative review, not a systematic review or meta-analysis.

*The discussion on MRI and ultrasound is descriptive but lacks critical evaluation (e.g., sensitivity, specificity, operator variability).

*While cadaveric data are valuable, the paper does not distinguish clearly between clinical applicability vs. anatomical observations.

*Some references are duplicated (e.g., Ngeow 2021 appears multiple times for different contexts).

Some suggestions to improve the paper:

  • Clinical decision algorithm for preoperative imaging based on anatomic risk factors.

  • Recommendations for training in ultrasound/MRI interpretation for oral surgeons.

Round 2

Reviewer 1 Report

Comments and Suggestions for Authors

I would like to once again congratulate the authors for their work titled “Challenges in Diagnosing the Course of the Lingual Nerve for Clinical Practice and Research.” Upon detailed evaluation of your revised manuscript, I observed that you have made commendable revisions. I also appreciate the fact that you have responded thoroughly to all of my comments. In particular, it is noteworthy that you created the figures yourselves, which is praiseworthy.

However, unfortunately, upon reviewing the iThenticate similarity report for the manuscript, I noticed a similarity index of 34%, which is unacceptably high for a scientific paper. This is quite disappointing. Nonetheless, your efforts are certainly appreciated. Therefore, I believe you should be given another opportunity to reduce the similarity index to below 20%.

I wish you the best of luck.

Author Response

Comment: I would like to once again congratulate the authors for their work titled “Challenges in Diagnosing the Course of the Lingual Nerve for Clinical Practice and Research.” 

Response: Thank you very much

Comment: Upon detailed evaluation of your revised manuscript, I observed that you have made commendable revisions. I also appreciate the fact that you have responded thoroughly to all of my comments. In particular, it is noteworthy that you created the figures yourselves, which is praiseworthy.

Response: the hard work and comprehensive review efforts with many valuable suggestions and recommendations to improve the manuscript are very much appreciated by all authors, thank you very much

Comment: However, unfortunately, upon reviewing the iThenticate similarity report for the manuscript, I noticed a similarity index of 34%, which is unacceptably high for a scientific paper. This is quite disappointing. Nonetheless, your efforts are certainly appreciated. Therefore, I believe you should be given another opportunity to reduce the similarity index to below 20%.

Response: As this is a review paper, many papers were quoted, perhaps this has caused the similarity index to be rather higher than normal, this is our mistake. However, we have attempted some changes as highlighted in the manuscript, although some technical terms are difficult to avoid similarity. Therefore we hope the current amendments can meet the requirements. Our latest recheck for similarity results shows 18% similarity.

Reviewer 2 Report

Comments and Suggestions for Authors

all commnets and suggestions were accepted.

Author Response

Comment: all comments and suggestions were accepted.

Response: the hard work and comprehensive review efforts with many valuable suggestions and recommendations to improve the manuscript are very much appreciated by all authors, thank you very much

Round 3

Reviewer 1 Report

Comments and Suggestions for Authors

I would like to once again congratulate the authors for their review titled “Challenges in Diagnosing the Course of the Lingual Nerve for Clinical Practice and Research.” I particularly appreciate that the figures were created by the authors themselves, which I find commendable. After the third round of review, I believe the manuscript is now ready for publication. Indeed, the reduction of the previously high plagiarism rate to 18% is also noteworthy and praiseworthy. In this context, I am confident that their review will contribute to the literature, and I congratulate the authors once more and wish them continued success.

Comments on the Quality of English Language

There may be a need for editing in terms of English grammar and language use. In this context, professional editing might be necessary.